# The Impact of Surgical Practice on Oncological Outcomes in Robot-Assisted Radical Hysterectomy for Early-Stage Cervical Cancer, Spanish National Registry

**DOI:** 10.3390/cancers14030698

**Published:** 2022-01-29

**Authors:** Sergi Fernandez-Gonzalez, Jordi Ponce, María Ángeles Martínez-Maestre, Marc Barahona, Natalia R. Gómez-Hidalgo, Berta Díaz-Feijoo, Andrea Casajuana, Myriam Gracia, Jon Frias-Gomez, Yolanda Benavente, Laura Costas, Lola Martí, Lidia Melero, Jose Manuel Silvan, Eva Beiro, Ignacio Lobo, Jesús De la Rosa, Pluvio J. Coronado, Antonio Gil-Moreno

**Affiliations:** 1Department of Gynecology, Bellvitge University Hospital, IDIBELL, L’Hospitalet de Llobregat, Universitat de Barcelona, 08907 Barcelona, Spain; mbarahona@bellvitgehospital.cat (M.B.); lmarti@bellvitgehospital.cat (L.M.); 2Department of Gynecology, Hospital Universitario Virgen del Rocio, 41001 Sevilla, Spain; mariaa.martinez.maestre.sspa@juntadeandalucia.es (M.Á.M.-M.); lidiam.melero.sspa@juntadeandalucia.es (L.M.); josem.silvan.sspa@juntadeandalucia.es (J.M.S.); 3Department of Gynecologic Oncology, Hospital Universitari Vall d’Hebron, 08035 Barcelona, Spain; nat.rodriguez@vhebron.net (N.R.G.-H.); agil@vhebron.net (A.G.-M.); 4Institute Clinic of Gynecology, Obstetrics and Neonatotlogy, Hospital Clinic, Institut d’Investigacions Biomèdiques August Pi I Sunyer (IDIBAPS), Universitat de Barcelona, 08036 Barcelona, Spain; bdiazfe@clinic.cat; 5Instituto de Salud de la Mujer (IdISSC), Hospital Clínico San Carlos, Universidad Complutense Madrid, 28040 Madrid, Spain; andreamercedes.casajuana@salud.madrid.org (A.C.); myriam.gracia@salud.madrid.org (M.G.); pluviojesus.coronado@salud.madrid.org (P.J.C.); 6Cancer Epidemiology Research Programme, Catalan Institute of Oncology, IDIBELL, L’Hospitalet de Llobregat, 08907 Barcelona, Spain; jfrias_ext@iconcologia.net (J.F.-G.); ybenavente@iconcologia.net (Y.B.); lcostas@iconcologia.net (L.C.); 7Department of Gynecology, Hospital Universitario de Basurto, 48013 Bilbao, Spain; eva.beirofelipe@osakidetza.eus (E.B.); ignacio.lobo@quironsalud.es (I.L.); jesushilario.delarosa@ehu.es (J.D.l.R.)

**Keywords:** early-stage cervical cancer, robotic surgery, radical hysterectomy, oncological outcome, recurrence, disease-free survival, surgical practice, surgery

## Abstract

**Simple Summary:**

Minimal invasive surgery (MIS) has been associated with lower disease-free survival than open surgery among women who underwent radical hysterectomy for early-stage cervical cancer. However, the mechanisms by which MIS increases mortality in cervical cancer remain uncertain. We aimed to determine if surgical practice among centers using robotic surgery has an impact on oncological outcomes. We evaluated 215 women with early-stage cervical cancer (≤IB1 or IIA1, FIGO 2009) who underwent robot-assisted radical hysterectomy in five Spanish tertiary centers between 2009 and 2018. A higher surgical volume, higher participation in clinical trials, higher rate of MRI use for diagnosis, greater use of sentinel lymph node biopsies, and a favorable learning curve with low rates of early recurrences were observed for the centers with better oncological outcomes. These factors might have a significant impact on oncological outcomes in all surgical approaches.

**Abstract:**

This study aimed to assess whether surgical practice had a significant impact on oncological outcomes among women who underwent robot-assisted radical hysterectomy for early-stage cervical cancer (≤IB1 or IIA1, FIGO 2009). The secondary objective was to audit the pre-surgical quality indicators (QI) proposed by the European Society of Gynaecological Oncology (ESGO). The top 5 of 10 centers in Spain and Portugal were included in the analysis. The hospitals were divided into group A (*n* = 118) and group B (*n* = 97), with recurrence rates of <10% and >10%, respectively. After balancing both groups using the propensity score, the ORs for all events were higher and statistically significant for group B (recurrences OR = 1.23, 95% CI = 1.13–1.15, *p*-value = 0.001; death OR = 1.10, 95% CI = 1.02–1.18, *p*-value = 0.012; disease-specific mortality OR_r_ = 1.11, 95% CI = 1.04–1.19, *p*-value = 0.002). A higher surgical volume, higher participation in clinical trials, higher rate of MRI use for diagnosis, greater use of sentinel lymph node biopsies, and a favorable learning curve with low rates of early recurrences were observed among the centers with better oncological outcomes. These factors might have a significant impact on oncological outcomes not only after robot-assisted surgery, but also after laparoscopies and open surgeries in the treatment of cervical cancer.

## 1. Introduction

The surgical approach for treating women with early-stage cervical cancer changed in favor of open surgery in 2018 after the publication of the LACC trial [1]. In that randomized study, the disease-free survival (DFS) rate at 4.5 years was lower for patients who underwent minimally invasive surgery (MIS) than for those subjected to open surgery (86.0% vs. 96.5%, respectively). In recent years, studies have focused on validating these results by comparing the open approach to MIS, with a recent meta-analysis producing the same findings [2]. The mechanisms by which MIS increases mortality in cervical cancer patients remain uncertain, and different theories have been postulated such as the spread of cancer cells from uterine manipulation or colpotomy and the effects of gas (CO_2_) insufflation. However, the European multicenter retrospective SUCCOR study observed a similar risk of relapse in patients affected by early-stage cervical cancer who had undergone MIS with protective surgical maneuvers when compared to open surgery [3].

The European Society of Gynaecological Oncology (ESGO) aims to homogenize the clinical and surgical practice in gynecological cancers. Accordingly, they published quality indicators (QIs) for treating cervical cancer in 2020 [4] to promote good clinical practice. Interestingly, the implementation of the QIs across Europe has been evaluated [5], and high rates of adjuvant therapy and low rates of sentinel lymph node detection have been observed. Consequently, to attain the best oncological outcomes, further research and enhanced surgical training are recommended.

New studies such as the RACC trial [6] have been designed to elucidate whether there is still a role for MIS in the treatment of cervical cancer. The randomized RACC trial aims to investigate the oncological safety of robotic radical hysterectomies in comparison with open surgery. All the theories mentioned above have been taken into consideration in the design of the study. However, recruitment is ongoing, and the investigation of the risk factors for recurrences in cervical cancer is still necessary. In this regard, the evolution of the radical hysterectomy technique [7,8,9,10,11,12] since its first description in 1912 [13] reflects the challenges that have been associated with this surgery. Very recently, a new classification of radical hysterectomy has been postulated considering the lateral extent of parametrium resection and the depth of resection of the resected vaginal vault without or with its three-dimensional paracolpium [14]. However, the impact of surgical practice on oncological outcomes after robot-assisted radical hysterectomies remains unclear.

Therefore, the primary objective of our study was to determine if surgical practice among centers using robotic surgery in Spain for early-stage cervical cancer has an impact on oncological outcomes. The secondary objective was to audit the pre-surgical QIs proposed by the ESGO.

## 2. Materials and Methods

### 2.1. Study Design and Participants

All centers that performed robot-assisted radical hysterectomies in Spain and Portugal in patients diagnosed with early-stage cervical cancer were eligible for initial inclusion (Table 1). We selected the hospitals with ≥20 patients who underwent surgery during the study period, and they were analyzed according to the recurrence rate. Centers with recurrence rates <10% formed group A, while those with recurrence rates >10% formed group B for analysis in a retrospective multicenter cohort study. The inclusion criteria were: infiltrating cervical cancer diagnosed from a biopsy, stage IA1–IIA1 based on the FIGO 2009 classification, and treatment with robotic surgery from the start of the robotic surgery programs up to 2018. Exclusion criterion was a FIGO 2009 stage ≥IB2 (except IIA1). All data were collected from 5 tertiary hospitals in Spain after anonymization. The primary objective of the present study was to assess whether the surgical practice had a significant impact on oncological outcomes. The secondary objective was to audit the pre-surgical results according to the recommendations of the quality indicators (QI) for cervical cancer proposed by the ESGO [4].

### 2.2. Surgery and Adjuvant Treatment

All radical hysterectomies were performed with the da Vinci^®^ Surgical System Version P9 (manufactured in Sunnyvale, CA, USA) and were classified according to the Querleu–Morrow classification [15]. Pelvic lymph node status was evaluated by means of SLNBs and/or pelvic lymphadenectomies, depending on the surgeon’s criterion. Cases of positive pelvic lymph nodes were all patients with positive lymph nodes that included micrometastases and isolated tumor cells after SLNB ultrastaging. SLNBs were processed with technetium ± blue dye or indocyanine green (ICG) or ICG alone. Macrometastases, micrometastases, and isolated tumor cells were analyzed by SLNB ultrastaging. When macrometastases were confirmed in frozen sections, a radical hysterectomy was not performed.

Adjuvant treatment included external beam radiotherapy (50 Gy/5 weeks) ± brachytherapy ± chemotherapy (40 mg/m^2^ of cisplatin every week during external beam radiation therapy). It was indicated depending on the protocol of the center for the management of cervical cancer, taking into account the FIGO stage, risk factors according to Sedlis criteria [15], and positive margins.

### 2.3. Oncological Outcomes and Audit of Quality Indicators

Recurrences were diagnosed by a combination of clinical, radiological, and histological findings. Time from surgery to diagnosis of the recurrence was used to define DFS. Time from surgery to death from any cause or from cervical cancer was applied to define OS and disease-specific mortality, respectively. The audit of oncological outcomes was performed by comparing the results obtained with the recommended targets of the QIs proposed by the ESGO [4]. Among the 15 QIs specifically designed for the analysis of surgical or treatment outcomes, 8 were selected.

### 2.4. Data Collection

The clinical and pathological variables studied included age, body mass index (BMI, kg/m^2^), clinical tumor size by inspection or ultrasound, size measured by MRI, tumor histology, FIGO stage after surgery, histological tumor size, surgical margins, presence of invasion of the lymphovascular space (LVSI), and tumor grade. LVSI was diagnosed when malignant cells were present in the epithelial tissue-lined spaces of the cervical stroma. The pathologist recorded the size of the cervical tumor as the largest diameter on the cone and gross tissue samples. Grade II–IV complications were recorded intraoperatively [16], during the first 30 postoperative days (early complications) and after 30 postoperative days (late complications) [17]. Ethical approval was not applicable due to the retrospective nature of this study.

### 2.5. Statistical Analysis

Inverse probability of treatment weighting (IPTW) using a propensity score (PS) [18] was used to balance the sample in groups A and B according to the following baseline covariates: age (continuous), body mass index (BMI) (<25 and ≥25), histology (squamous, adenocarcinoma, and others), size (<20 and >20 mm), grade (grade 1 and grade 2–3), lymphovascular invasion (yes and no), adjuvant treatment (yes and no), and nodal status (negative, positive, and not assessed). We performed unadjusted logistic regression to estimate the odds ratios (ORs) and 95% confidence intervals (95% CI) for three outcomes: recurrence, death from any cause, and death from cervical cancer using the synthetic balanced sample. Differences in the distribution of potential factors that might distinguish group A from group B were assessed by the chi-square test for categorical variables and the Wilcoxon rank-sum test for normally distributed and non-normally distributed continuous variables. Categorical variables are reported in absolute numbers and percentages. Mean values and standard deviation (SD) or median values and range (minimum–maximum) were used to express the continuous variables. DFS was calculated with the Kaplan–Meier method, with differences in the probability of survival analyzed with the log-rank test. The date of the last follow-up was the censored date for patients without events. The significance level was established at 0.05, and all statistical tests were two-sided. For analyses and plots, we used IBM SPSS version 25.0 and Stata 16 (College Station, TX, USA: StataCorp LP; teffects function with IPTW option).

## 3. Results

A total of 239 women with a clinical diagnosis of early-stage (IA1, IA2, IB1, or IIA1) cervical cancer underwent robot-assisted radical hysterectomy during the study period. Of these women, 215 (90%) were treated in 5 tertiary care hospitals. A target of recurrence rate <10% vs. >10% was applied, according to the QI of the ESGO [4], in order to divide the hospitals into group A (*n* = 118) and group B (*n* = 97), respectively (Table 1). The rates of disease-free survival (DFS) at 5 years were 94.7% and 77.4% in groups A and B, respectively, with a hazard ratio (HR) of 4.43 (95% confidence interval [CI] = 1.65–11.89; *p* = 0.003) (Appendix A).

### 3.1. Clinical and Pathological Characteristics

Table 2 summarizes the clinical and pathological characteristics of the patients before and after propensity score weighting. Patients in group A were significantly more likely to have larger tumors and higher rates of lymphovascular invasion and positive lymph nodes. These covariates, in addition to others considered to show differences that were of near significance, were well balanced after the inverse probability of treatment weighting (IPTW), with a *p*-value of 0.9483.

### 3.2. Oncological Outcomes

In the propensity score-weighted cohort, 5 (4.3%) vs. 19 (19.6%) recurrences were observed in group A and group B, respectively, which corresponded to a 23% higher odds ratio of recurrence (OR = 1.23; 95% CI = 1.13–1.35; *p*-value = 0.001). A significantly higher odds ratios of overall mortality and cervical cancer-associated mortality were also observed in group B compared to group A after 5 years of follow-up (Figure 1).

### 3.3. Audit of the Quality Indicators

Five of the eight QIs of the ESGO for the treatment of cervical cancer [4] were fulfilled by both groups. The minimum number of radical hysterectomies per year and the rate of pre-operative assessment including MRI assessments were the two QIs that were not fulfilled in both groups (Table 3). The QI of clinical trial participation was only fulfilled by group A.

### 3.4. Surgical Assessment of the Two Groups

Additional indicators of surgical activity in the two groups are detailed in Table 4. We observed that the number of robotic surgery procedures was 40.1 and 32.2 in group A and B, respectively. It indicated 25% [8] more surgeries performed in group A. Longer surgical times were observed in group A vs. B (258.6 vs. 221.9 min, respectively; *p* < 0.001). Additionally, Appendix A presents the learning curves in surgical practice in the two groups. Sentinel lymph node biopsies (SLNB) were performed in 88/118 (74.6%) vs. 11/97 (11.3%) (*p* < 0.001) of the patients in groups A and B, respectively. There were no significant differences between the groups in the rate of pelvic lymphadenectomies performed and in the mean number of lymph nodes excised (Table 4). An intrauterine manipulator was used in 98/118 (83.1%) vs. 55/97 (56.7%) of the patients (*p* < 0.001) in groups A and B, respectively. Intra-operative and post-operative complications were observed in 1 vs. 6 patients (*p* = 0.028) and in 6 vs. 12 patients (*p* = 0.055), in groups A and B, respectively (Appendix A). The total number of radical hysterectomies performed per center and year is detailed in Appendix A.

A proportional relationship was observed between the number of MRI assessments not performed before surgery and recurrences (Figure 2).

In group B, 8 out of the 34 patients who had undergone surgery in the first 2 years after the introduction of the robotic program relapsed. Six of them relapsed before the two-year follow-up (early recurrences). In group A, among the 19 patients who had undergone surgery in the first 2 years after the introduction of the robotic program, 1 relapsed.

## 4. Discussion

Studies published in the last years on the surgical treatment of early-stage cervical cancer indicate that an open radical hysterectomy is safer than minimally invasive surgery. Therefore, the latter should be exclusively performed under investigation in clinical trials. However, the opportunity to analyze real-world data from all radical hysterectomies performed to date with robot-assisted approaches must be taken advantage of. To our knowledge, this is a novel study for analyzing the impact of surgical practice and the factors associated with better oncological outcomes among patients who underwent surgery by the same route. IPTW using a propensity score was applied to balance patients between group A (centers with recurrence rates <10%) and group B (centers with recurrence rates >10%) according to age, BMI, histology, size, tumor grade, lymphovascular invasion, adjuvant treatment, and nodal status. The ORs for all events were higher and statistically significant for group B when compared to group A (OR_Recurrence_ = 1.23, 95% CI = 1.13–1.15, *p*-value = 0.001; OR_Death_ = 1.10, 95% CI = 1.02–1.18, *p*-value = 0.012; and OR_Death-cervical-cancer_ = 1.11, 95% CI = 1.04–1.19, *p*-value = 0.002). Therefore, these differences were not related to clinical or tumoral factors and were attributed to differences in surgical practice between the groups. A higher surgical volume, higher participation in ongoing clinical trials, higher rate of MRI assessments for diagnosis, greater use of SLNBs, and a favorable learning curve with low rates of early recurrences were observed among centers with better oncological outcomes.

Thanks to the effort made firstly by the European Organisation for Research and Treatment of Cancer (EORTC) [20] and recently by the ESGO [4] in defining the factors that influence the quality of radical hysterectomies, all institutions or cooperating groups can evaluate their results to improve their clinical practice. Surgical volume is one of the most studied surgical indicators [21], and the optimal target is ≥30 radical procedures (in cervical cancer) per year according to ESGO recommendations. However, none of our groups met even the minimum target, which is ≥15 radical procedures. A low prevalence of cervical cancer in western Europe and the lack of centralization compared to Nordic countries [22] might explain these results. Interestingly, the cutoff for surgical volume related to DFS in cervical cancer was examined in a recent study published by the Japanese Gynecologic Oncology Group (JGOG) [23] that analyzed 5964 radical hysterectomies from 649 institutions in Japan. High-volume centers were defined as those performing ≥105 radical hysterectomies over 5 years. The study observed lower recurrence rates and lower mortality in women treated by high-volume institutions. However, we observed a DFS rate of 94.7% (DFS) at 5 years for the centers in group A (*n* = 118). We hypothesize that the high-quality clinical practice would balance out the effect of the low surgical volume.

Our data also showed a relationship between the learning curve and the recurrence rate, which is presented in Figure 3. Interestingly, the initial effects of adopting robotic surgery were recently studied in a retrospective evaluation of 2202 cervical cancer cases [24], which revealed that a higher hospital surgical volume was associated with an increased rate of perioperative complications after robot-assisted radical hysterectomies and a decreased rate of complications after traditional laparoscopies or laparotomies. These findings were attributed to the recent introduction of a new surgical device in comparison with traditional approaches such as open surgery. However, these data were extracted from ICD-9 codes without a review of the medical records. Consequently, the authors of the study concluded that an analysis of the confounders should have been performed. By contrast, our data were reviewed by surgeons with expertise in gynecologic oncology, and all five centers evaluated in our study started robotic surgery in 2009. Consequently, in the present study, the learning curve between the centers was evaluated with a minimum risk of bias. Surprisingly, despite all the centers starting robotic surgery at the same time, we observed two different trends in early recurrences (<2 years of follow-up) when comparing groups A and B. Figure 3 shows no vs. six early recurrences, respectively, in the first two years after the introduction of the robotic program. Therefore, considering that the ratio of the number of radical hysterectomies to the number of surgeons was 4.4 and 4.3 in groups A and B, respectively, we propose that the quality of the surgery might be as important as the quantity. The lower rates of complications observed in group A (Appendix A) support this hypothesis.

In contrast to the findings of other studies [3], our data indicated that the use of an intrauterine manipulator is not associated with worse clinical outcomes [25]. The application of protective maneuvers may partly explain some of the recurrences in patients who undergo MIS. Several protective factors against recurrences have been postulated, such as cervical conization prior to a radical hysterectomy [26,27]. Consequently, many aspects of the current clinical practice might explain the results of the LACC trial. However, considering that these new findings were drawn from retrospective studies, further research is necessary to endorse them. In this regard, a national German survey showed that wrong surgical principles might be the second cause explaining worse prognosis in the MIS group [28]. For this reason, we aimed to evaluate the indicators related to expertise in gynecologic oncology among all the centers included in our study. Interestingly, in all centers, radical hysterectomies were performed/supervised by a qualified surgeon, and treatment was discussed by a multidisciplinary team. However, the centers in group A had a greater involvement in ongoing clinical trials compared to those in group B (≥5 trials vs. <1, respectively). These differences might explain the fact that SLNBs were performed in 74.6% vs. 11.3% of the patients in groups A and B, respectively. Indeed, the Senticol [29] and Sentix [30] trials were open during the study period. The performance of SLNBs explains the longer mean time of surgery observed in group A when compared to group B (258.6 vs. 221.9 min, respectively). In our study population, ultrastaging allowed the detection of three micrometastases (mi) and three isolated tumor cells (itc) in group A and 0 mi/itc in group B. While a recent well-designed retrospective study observed a significantly decreased DFS in patients affected by micrometastases [31], an ancillary study involving patients from the Senticol 1 and 2 trials did not show this association [32]. Therefore, the impact of micrometastases on the prognosis of patients affected by early-stage cervical cancer remains unclear.

Despite our analysis being retrospective, the specific objectives in addition to the good quality of the study design may overcome selection bias. Accordingly, we balanced both groups using propensity score matching to reduce the bias of heterogeneity and confounders. The sample size of our work might be considered smaller than that of other studies carried out in countries with higher prevalence of cervical cancer and higher centralization. However, the number of events (death or recurrence) in our population allowed to analyze our objectives. Finally, heterogeneity bias was reduced by selecting all consecutive eligible women undergoing robotic surgery for early-stage cervical cancer. Finally, bias deriving from the starting up of the robotic program might have had an equal impact in both groups. This novel approach in such a difficult topic may encourage further investigations on surgical skills or treatment strategies in different centers.

## 5. Conclusions

In conclusion, we observed significant differences in the recurrence rate between tertiary care centers that performed robot-assisted radical hysterectomies for early-stage cervical cancer in the same period of time. The differences in recurrence rate, overall mortality, and disease-specific mortality remained significant after statistical balancing according to clinical covariates and adjuvant treatment. Our results indicated that centers with better oncological outcomes were associated with higher surgical volume, but also higher rates of preoperative MRI assessments, greater participation in ongoing clinical trials, higher rates of SLNBs, and favorable learning curves with low early recurrences. Consequently, these factors might have a significant impact on oncological outcomes not only after robot-assisted surgery, but also after laparoscopies and open surgeries in the treatment of cervical cancer.

## Figures and Tables

**Figure 1 cancers-14-00698-f001:**
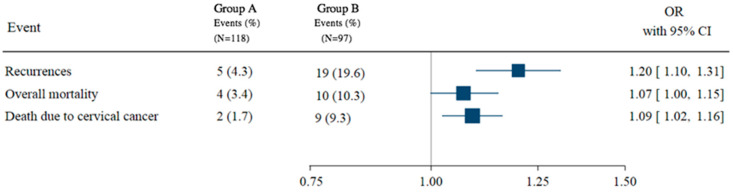
Oncological outcomes after balancing group A and group B with the IPTW. The odds of recurrence, overall mortality, and disease-specific mortality after balancing the sample for age, BMI, histology, size, grade, lymphovascular invasion, adjuvant treatment, and nodal status. ORs for all events were higher and statistically significant for group B compared to group A (OR_Recurrence_ = 1.23, 95% CI = 1.13–1.15, *p*-value = 0.001; OR_Death_ = 1.10, 95% CI = 1.02–1.18, *p*-value = 0.012; OR_Death-cervical-cancer_ = 1.11, 95% CI = 1.04–1.19, *p*-value = 0.002).

**Figure 2 cancers-14-00698-f002:**
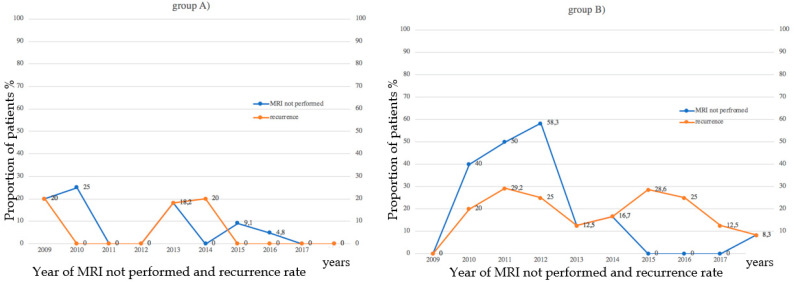
Proportional relationship between pre-surgical MRI assessments and the recurrence rate. A higher recurrence rate per year was observed in both groups when pre-operative MRI assessments were not performed.

**Figure 3 cancers-14-00698-f003:**
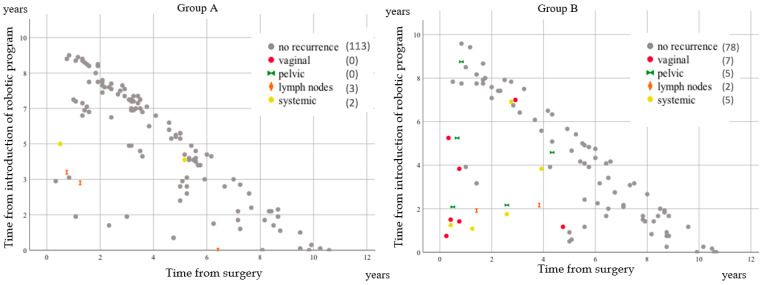
Relationship between the time of recurrence and the time from the introduction of the robotic program. All recurrences (in color) and no events (in gray) in both groups are presented on scatter graphs. Five and 19 recurrences in groups **A** and **B**, respectively, are presented in relation to the follow-up period and the time from the introduction of the robotic program.

**Table 1 cancers-14-00698-t001:** Distribution of patients and recurrence by centers.

Items	Patients	Recurrences (%)		
Center (a)	53	1 (1.9)	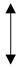	Group A
Center (b)	46	3 (6.5)
Center (c)	19	1 (5.3)
Center (d)	55	13 (23.6)	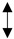	Group B
Center (e)	42	6 (16.7)
Center (f)	9	-		
Center (g)	6	-		
Center (h)	5	-		
Center (i)	2	-		
Center (j)	2	-		
Total	239	24		

**Table 2 cancers-14-00698-t002:** Baseline characteristics of the patients in groups A and B in the original and the balanced samples.

Items	Group A (*n* = 118)	Group B (*n* = 97)	*p*-Value	Group A (*n* = 118)	Group B (*n* = 97)	*p*-Value *
Age (mean, sd)	47.0 ± 10.8	49.7 ± 10.7	0.068	49.1 ± 11.0	48.8 ± 10.5	0.839
BMI (%)						
<25	51 (43.6)	36 (37.1)	0.337	48 (40.2)	39 (40.2)	0.996
≥25	66 (56.4)	61 (62.9)		70 (59.8)	58 (59.8)	
Histology (%)						
Squamous cell carcinoma	70 (59.3)	61 (62.9)	0.274	73 (61.6)	61 (62.9)	0.927
Adenocarcinoma	45 (38.1)	30 (30.9)		41 (35)	32 (33)	
Others	3 (2.5)	6 (6.2)		4 (3.4)	4 (4.1)	
Size (%)						
<20 mm	68 (57.6)	70 (72.2)	0.027	77 (65)	62 (63.9)	0.874
>20 mm	50 (42.4)	27 (27.8)		41 (35)	35 (36.1)	
Grade (%)						
Grade 1	31 (26.3)	34 (35)	0.163	40 (33.3)	30 (30.9)	0.708
Grade 2–3	87 (73.7)	63 (65)		78 (66.7)	67 (69.1)	
Lymphovascular invasion (%)						
No	89 (75.4)	86 (88.7)	0.013	96 (81.2)	78 (80.4)	0.885
Yes	29 (24.6)	11 (11.3)		22 (18.8)	19 (19.6)	
Adjuvant treatment (%)						
No	83 (70.3)	74 (76.3)	0.328	86 (72.7)	70 (72.2)	0.937
Yes	35 (29.7)	23 (23.7)		32 (27.3)	27 (27.8)	
Nodal status (%)						
Negative	108 (91.5)	86 (88.7)	0.014	104 (88.0)	87 (89.7)	0.731
Positive	8 (6.8)	2 (2.1)		5 (4.3)	5 (5.2)	
Not assessed	2 (1.7)	9 (9.3)		9 (7.7)	5 (5.2)	

Chi-square test or Fisher’s exact test for the comparison of categorical variables and Student’s *t* test for continuous variables; *p*-value *: after balancing with the propensity score.

**Table 3 cancers-14-00698-t003:** ESGO quality indicators related to the treatment of cervical cancer.

ESGO Quality Indicator	Target Value	Group A	Group B
QI 1-Number of radical procedures (parametrectomies) in cervical cancer performed per center per year	≥15	8.9	6.7
QI 2-Surgery performed or supervised by a certified gynecologic oncologist or a trained surgeon dedicated to gynecological cancers	100%	100%	100%
QI 3-Center participating in ongoing clinical trials in gynecological cancers	≥1	≥5	<1
QI 4-Treatment discussed at a multidisciplinary team meeting	100%	100%	100%
QI 5-Required pre-operative investigation	100%	92.5%	80.7%
QI 6-Minimum required elements in surgical reports	100%	100%	100%
QI 7-Minimum required elements in pathology and pathology reports	≥ 90%	100%	100%
QI 8-Structured prospective reporting of follow-up and 30-day post-operative morbidity	≥90%	100%	100%

**Table 4 cancers-14-00698-t004:** Surgical activity and surgical techniques.

Indicators	Group A	Group B	*p*-Value
Number of robotic procedures per center per year	40.1	32.2	0.0400
Ratio of the number of robotic radical hysterectomies to the number of surgeons per year	4.4	4.3	0.850
Surgeons in gynecologic oncology per center	4.3	5	0.079
Surgical time, minutes (mean ± SD)	258.6 ± 51.8	221.9 ± 70.9	<0.001
Sentinel lymph node biopsy performed (%)	88/118 (74.6)	11/97 (11.3)	<0.001
Pelvic lymphadenectomies performed (%)	109/118 (92.4)	85/97 (87.6)	0.258
Right pelvic lymph nodes, median (range)	9 (1–26)	9 (1–21)	0.840
Left pelvic lymph nodes, median (range)	8 (3–27)	7 (2–24)	0.424
Type of radical hysterectomy *, *n* (%)			0.203
A	0	12/97 (12.4)	
B1	34/118 (28.8)	17/97 (17.5)	
B2	3/118 (2.5)	6/97 (6.2)	
C1	81/118 (68.6)	62/97(63.9)	
Clear surgical margins (%)	117/118 (99.2)	93/97 (95.9)	0.113
Intrauterine manipulator, *n* (%)	98/118 (83.1)	55/97 (56.7)	<0.001
Nerve sparing technique, *n* (%)	118/118 (100)	96/97 (99)	0.451
Hospital stay (in days), median (range)	3 (1–21)	3 (1–13)	0.478

Type of radical hysterectomy *: according to the Querleu–Morrow classification [19]; chi-square test or Fisher’s exact test for the comparison of categorical variables and Student’s *t*-test for continuous variables.

## Data Availability

The data presented in this study are available from the corresponding author. The data are not publicly available due to privacy.

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
