# Peer review of "The Impact of Surgical Practice on Oncological Outcomes in Robot-Assisted Radical Hysterectomy for Early-Stage Cervical Cancer, Spanish National Registry"

_cancers, 2022, doi:10.3390/cancers14030698_

Round 1

Reviewer 1 Report

I was particularly pleased to review the manuscript titled "The impact of surgical practice on oncological outcomes in robot-assisted radical hysterectomy for early-stage cervical cancer. Spanish National Registry”.

The authors performed a retrospective analysis of the Spanish Regional Registry to analyze the role of surgical practice in the oncological outcomes of early-stage cervical cancer patients undergoing robotic-assisted surgery. The manuscript's topic is of high interest, and the manuscript is in line with the Journal’s aims.

The study was well structured, and the paper is well written.

However, I have some major concerns.

Given that the topic is intensely debated, it is hard to drive significantly enriching conclusions from a retrospective analysis reporting data from 239 patients treated in 5 different hospitals during the decade between 2009 and 2018.

However, it is interesting and of undisputed scientific value how the Authors retrospectively applied the ESGO Quality Indicators, further underlining the importance of the correct management of cervical cancer delivered by high volume centers: this is what has actually been done in the Hospitals in Group A.

However, the Authors should widely discuss the limitations of their study:

-      Small population of patients

-      Bias deriving from the starting up of the robotic program

-      Bias deriving from the heterogeneity of Centers and the low volume/year

In addition, since the ESGO in a statement on the laparoscopic radical hysterectomy suggested avoiding any maneuver that can determine spillage of tumor cells (including the uterine manipulator), the Authors should argue in moderation results regarding the use of intrauterine manipulator in their study. It is a retrospective analysis up to 2018: would they still use the uterine manipulator today?

The section M&M needs to be moved following the Intro. 

Minor concerns

Line 313: How many Centers have been evaluated for eligibility, and how many were considered eligible?

Line 315: “we considered Hospitals with ≥ 20” …per year?

Line 117: “A target of recurrence 117 rates < 10% was applied according to the QI of the ESGO (4) to divide the hospitals into 118 group A (n = 118) and group B (n = 97), which had recurrence rates of < 10% and > 10%, 119 respectively”: This sentence is unclear. If the Author “divided the Hospital” into two groups they should indicate the number of Hospitals first, and then how many patients operated in each group.

Line 171: there is one “respectively” more

Author Response

The authors appreciate very much your supportive comments which have contributed to improve the quality of the manuscript.

REVIEWER 1

However, the Authors should widely discuss the limitations of their study:

-      Small population of patients

-      Bias deriving from the starting up of the robotic program

-      Bias deriving from the heterogeneity of Centers and the low volume/year

The original article has been modified in lines 321-330 according to these observations:
“Accordingly, we balanced both groups using propensity score matching to reduce the bias of heterogeneity and confounders. The sample size of our work might be considered smaller than other studies carried out in other countries with higher prevalence of cervical cancer and higher centralization. However, the number of events (death or recurrence) in our population allows to analyze our objectives. Finally, heterogeneity bias was reduced by selecting all consecutive eligible women undergoing robotic surgery for early-stage cervical cancer. Finally, bias deriving from the starting up of the robotic program might have an impact in both groups equally. This novel approach in such a difficult topic may encourage further investigations on surgical skills or treatment strategies between different centers.

In addition, since the ESGO in a statement on the laparoscopic radical hysterectomy suggested avoiding any maneuver that can determine spillage of tumor cells (including the uterine manipulator), the Authors should argue in moderation results regarding the use of intrauterine manipulator in their study. It is a retrospective analysis up to 2018: would they still use the uterine manipulator today?

We do agree about avoiding those maneuvers that can disseminate tumor cells as intrauterine manipulator. However, we observed a high rate of intrauterine manipulator use in patients who underwent surgery in group A centers. Therefore, we highly discussed our findings compared to literature as you can see from line 280 to 287

“In contrast to the findings of other studies (3), our data indicated that the use of an intrauterine manipulator is not associated with worse clinical outcomes (20). The application of protective maneuvers may partly explain some of the recurrences in patients who undergo MIS. Several protective factors against recurrences have been postulated, such as cervical conization prior to a radical hysterectomy (21)(22). Consequently, many aspects of current clinical practice might explain the results from the LACC trial. However, considering that these new findings are drawn from retrospective studies, further research is necessary to endorse them.”

The question of the reviewer: would they still use the uterine manipulator today?
In our opinion, this answer it is out of study period and it depends on each hospital.

The section M&M needs to be moved following the Intro. 
According to instructions for authors of Cancers, the section M&M needs to be written before discussion.

Minor concerns

Line 313: How many Centers have been evaluated for eligibility, and how many were considered eligible?

All centers that performed radical hysterectomy for early-stage cervical cancer in Spain and Portugal were included. In total, there were 10 hospitals as you can see in Figure 1. Among them, only centers that performed ≥ 20 radical hysterectomies were selected for this study.

Line 315: “we considered Hospitals with ≥ 20” …per year? In total.

A proper clarification has been added from line 337 to 338:
“All the centers that performed robot-assisted radical hysterectomies in Spain and Portugal in patients diagnosed with early-stage cervical cancer were eligible for initial inclusion (Figure 1). We selected the hospitals with  20 patients who underwent surgery during the study period and they were analyzed according to the recurrence rate”.

Line 117: “A target of recurrence 117 rates < 10% was applied according to the QI of the ESGO (4) to divide the hospitals into 118 group A (n = 118) and group B (n = 97), which had recurrence rates of < 10% and > 10%, 119 respectively”: This sentence is unclear. If the Author “divided the Hospital” into two groups they should indicate the number of Hospitals first, and then how many patients operated in each group.

Thank you for your clarification. A proper modification has been added in line 118 and 119.
“Of these women, 215 (90%) were treated in 5 tertiary care hospitals. A target of recurrence rate < 10% vs >10% was applied, according to the QI of the ESGO (4), in order to divide the hospitals into group A (n = 118) and group B (n = 97) respectively (Figure 1).”

Line 171: there is one “respectively” more.
It has been corrected. Thank you.

Reviewer 2 Report

Thank you for submitting this very interesting and good written study to Cancers and for giving me the opportunity to review this paper. I still have 2 suggestions:
1- Can you provide us please with all numbers of radical hysterectomies performed in these centers over the study period, or all radical hysterectomies were performed with robotic surgery?

2- How many centers participate now in the ongoing RACC-trial? I think this will be a good indicator to show, that the centers in the group A were better qualified to be allowed to participate in this trial with strict criteria for quality of surgery.

3- Page 3, line 106: citing the new published technique of nerve-sparing radical hysterectomy and the new classification of radical hysterectomy depending on the precise understanding of the anatomy of paracolpium and inferior hypogastric plexus is recommended here (Both publicatiens were published in Cancers too):

  • Muallem, M.Z. A New Anatomic and Staging-Oriented Classification of Radical Hysterectomy. Cancers 2021, 13, 3326. https://doi.org/10.3390/cancers13133326

Author Response

The authors appreciate very much your supportive evaluation which has contributed to improve the quality of the manuscript.

1- Can you provide us please with all numbers of radical hysterectomies performed in these centers over the study period, or all radical hysterectomies were performed with robotic surgery?
This question has been properly answered in line 182 to 184:
“The total number of radical hysterectomies performed per center and year is detailed in table S2 (Supplementary Material).”

And Table S2 has been added in Supplementary Material:

Table S2. Number of total radical hysterectomies per center.

Center\year

2009

2010

2011

2012

2013

2014

2015

2016

2017

2018

a)

11

12

11

7

11

3

7

13

5

3

b)

12

10

12

6

10

8

10

10

10

13

c)

5

4

5

5

4

5

4

6

5

8

d)

0

4

17

9

5

11

7

5

13

16

e)

5

3

8

6

6

5

5

3

2

4

2- How many centers participate now in the ongoing RACC-trial? I think this will be a good indicator to show, that the centers in the group A were better qualified to be allowed to participate in this trial with strict criteria for quality of surgery.

Dear author, this topic has been evaluated in the current study (table 2, QI3). Certainly, 2/3 centers in group A have been involved in RACC trial but 0/2 centers in group B. However, this information is confidential and it has been already evaluated in table 2 which it reflects your observation.

3- Page 3, line 106: citing the new published technique of nerve-sparing radical hysterectomy and the new classification of radical hysterectomy depending on the precise understanding of the anatomy of paracolpium and inferior hypogastric plexus is recommended here (Both publicatiens were published in Cancers too):

 We really appreciate your comment. The original article has been modified accordingly in line 112-115:
“Very recently, a new classification of radical hysterectomy has been postulated considering either lateral extent of parametrium resection and depth resection of the resected vaginal vault without or with its three-dimensional paracolpium (14). However, the impact of surgical practice on oncological outcomes after robot-assisted radical hysterectomies remains unclear.”

(14) reference has been added. (Muallem, M.Z. A New Anatomic and Staging-Oriented Classification of Radical Hysterectomy. Cancers 2021, 13, 3326.)

Round 2

Reviewer 1 Report

Dear Authors, thank you for your answers and for editing the manuscript in accordance with the major concerns previously outlined.